# Feasibility and acceptability of GeneXpert MTB/XDR implementation among healthcare workers in three low-middle income African countries

Sara Keller[1], Kogieleum Naidoo[2,3], Medhane Zekarias[4], Sussan Israel-Isah[5], Mohammed Shaka[4], Gcinile Gule[2,3], Anushka Naidoo[2,3], Miriam Bathnna[5], Judith Nomthandazo Dlamini-Miti[6], Kalkidan Yae[4], Evaezi Okpokoro[5], Alash'le Abimiku[5,7], Ahmed Bedru[4], Edine W. Tiemersma[1]* on behalf of the TRiAD Study Consortium¶

1 KNCV Tuberculosis Foundation, Den Haag, The Netherlands, 2 Centre for the Aids Programme of Research in South Africa, Durban, KwaZulu-Natal, South Africa, 3 SAMRC-CAPRISA HIV-TB Pathogenesis and Treatment Research Unit, Doris Duke Medical Research Institute, University of KwaZulu-Natal, Durban, KwaZulu-Natal, South Africa, 4 KNCV Tuberculosis Foundation, Ethiopia Office, Addis Ababa, Ethiopia, 5 International Research Centre of Excellence- Institute of Human Virology, Abuja, Nigeria, 6 University of the Witwatersrand, Johannesburg, South Africa, 7 University of Maryland School of Medicine Institute of Human Virology, Baltimore, Maryland, United States of America

¶ The TRIAD full membership list can be found in the Acknowledgments section
* edine.tiemersma@kncvtbc.org

## Abstract

### Background

Xpert MTB/XDR (Xpert-XDR) testing can significantly shorten time to initiating appropriate drug-resistant tuberculosis (DR-TB) treatment, but its introduction may impact laboratory workflow, especially in laboratories not currently performing drug susceptibility testing. This study evaluated the feasibility and acceptability of implementing the Xpert-XDR for rapid triage and selection of all-oral regimens for DR-TB.

### Method

This was a multi-country, multi-site qualitative study conducted between July and November 2023, as part of the larger TriAD (Triage test for All oral DR TB drugs) study implemented in South Africa, Ethiopia, and Nigeria. We conducted semi-structured in-depth interviews with clinicians, nurses and laboratory staff at each study site until thematic saturation was achieved. Additionally, we interviewed policy makers (n = 9) and people with TB (PWTB) (n = 11), to provide additional insight on the implementation of this new diagnostic assay.

### Results

Healthcare workers (n = 61) found the new workflow feasible and acceptable. It was the increased speed in which PWTB would receive a correct diagnosis and appropriate treatment that provided the biggest benefit to moving to Xpert-XDR for healthcare

**Data availability statement:** The qualitative data collected for this study contains sensitive information that could lead to the identification of patients, staff and policy makers, despite the data having been de-identified. Data access requests can be made: • For South Africa, Caprisa sites, via the Biomedical Research Ethics Committee of the University of Kwazulu-Natal at the following e-mail address: BREC@ukzn.ac.za, citing protocol reference number BREC/00002654/2021; for the University of Witwatersrand via the Human Research Ethics Committee at the following email address: EthicsRegulatory@witshealth.co.za, citing protocol reference number 210805; • For Nigeria via the National Health Research Ethics Committee of Nigeria at the following email address: secretary@nhrec.net, citing NHREC approval number NHREC/01/01/2007-01/06/2023; • For Ethiopia via the Ethiopian Public Health Institute Institutional Review Board at the following email address: info@ephi.gov.et, citing the EPHI-IRB and protocol reference number EPHI-IRB-384-2021.

**Funding:** This publication was produced by the TRiAD study which is part of the EDCTP2 program supported by the European Union (Grant RIA2019-2888-TRIAD).

**Competing interests:** The authors have declared that no competing interests exist.

workers and PWTB. Laboratory staff mentioned that Xpert-XDR had expedited and simplified the laboratory workflows. Role-appropriate and ongoing training is a key factor in effective implementation as described by policy makers and healthcare workers alike.

Barriers impacting the ability to perform Xpert-XDR included unstable power supply, internet, and temperature control. Additionally, the Xpert MTB/Rif Ultra test has higher sensitivity for the detection of TB than the Xpert-XDR test, leading to discordant test results.

## Conclusion

This study showed that implementation of Xpert-XDR in health facilities is both feasible and acceptable by all types of healthcare workers. Some barriers with Xpert-XDR are not exclusive to this particular diagnostic tool but are important to address when policy makers are deciding which tools to implement.

## Introduction

Sub-Saharan Africa carries 25% of the total burden of drug-resistant (DR-TB) [1]. Successful management of DR-TB is hindered by difficulties in both diagnosis and treatment, fueling ongoing transmission [2]. In East Africa, the pooled prevalence of multidrug-resistant tuberculosis (MDR-TB) among newly diagnosed and previously treated tuberculosis (TB) patients were 4% and 21% respectively [3]. Meanwhile, South Africa and Nigeria account for nearly 42% of the DR-TB cases in Africa [1,4].

Most settings follow a multi-step testing cascade involving initial low-complexity molecular WHO-recommended rapid diagnostic assays for the detection of TB, with or without rifampicin resistance. A widely used test is the Xpert MTB/RIF or Xpert MTB/RIF Ultra (Xpert-Ultra) assay which detects *Mycobacterium tuberculosis* (MTB) in sputum and simultaneously detects whether there is resistance to rifampicin [5]. Those whose tests identify rifampicin resistance undergo further testing to detect additional drug resistance using culture and phenotypic drug sensitivity testing (pDST) and/or Line Probe Assays (LPA) [5].

The Xpert MTB/XDR (Xpert-XDR) assay was recently introduced, which allows for the rapid and accurate detection of resistance to isoniazid, fluoroquinolones, ethionamide, capreomycin, kanamycin and amikacin. Prior to this the interval between diagnosis of rifampicin-resistant TB (RR-TB) and availability of the strain's susceptibility pattern against second-line drugs was usually 1–2 weeks and could take up to several months [6,7]. Moreover, the drug susceptibility tests used are usually only available at higher referral levels which can be physically distant from the clinic where sputum samples are collected and treatment is carried out [8,9]. As a result of this sub-optimal testing cascade, people with TB (PWTB) may receive a less effective regimen or a late adjustment of their treatment to include only effective drugs. This increases the risk of amplifying drug resistance, ongoing transmission of DR-TB, prolonged morbidity, and mortality [2,10], and highlights an essential need for novel

approaches that incorporate improved diagnostic accuracy and replace lengthy phenotypic diagnostic methods enabling timely initiation of treatment and treatment response monitoring [11].

TriAD (Triage test for All oral DR TB drugs) introduced Xpert-XDR testing as part of a rapid workflow for the selection of appropriate all-oral DR-TB regimens in three Sub-Saharan African countries: Ethiopia, Nigeria, and South Africa. Implementation of Xpert-XDR testing may have significant impact on the current workflow that exists in laboratories, especially in peripheral laboratories where there is currently no alternative option to pDST. It also has potential impact on nurses and clinicians as the proportion of PWTB who need treatment adjustments may decline [12].

Successful implementation of an intervention depends on its feasibility and acceptability to both intervention recipients and those providing the diagnosis and care. By exploring implementation of the Xpert-XDR assay from the perspective of those in the healthcare settings, future areas for improvement may be identified. To fit an intervention to its user's needs and thus improve acceptability; experiences, thoughts, believes and opinions of the users should be determined.

The TriAD study aims to evaluate the effectiveness, feasibility, acceptability and cost-effectiveness of implementing a rapid triaging algorithm with the use of Xpert-XDR and selection of all-oral regimens for DR-TB in South Africa, Ethiopia, and Nigeria [13]. To assess the programmatic and operational aspects of introducing this algorithm, we assessed its feasibility and acceptability, using the theoretical framework of acceptability of healthcare interventions developed by Sekhon et al. [14]. We included barriers and facilitators from the perspective of healthcare workers, policy makers and PWTB in South Africa, Ethiopia, and Nigeria.

## Methods

### Design and setting

This was a multi-country, multi-site qualitative study utilizing in-depth interviews (IDIs). The specific aim was to evaluate the context in which rapid triaging with the use of Xpert-XDR is implemented and executed as experienced by healthcare workers, PWTB and policymakers. In total, 13 clinical sites were selected; two in South Africa, four in Nigeria, and seven in Ethiopia. Purposive and convenience sampling methodologies were used. In Ethiopia, one site was excluded due to an existing security concern. Study sites varied in terms in their process of sample collection for running the assay. Table 1 specifies the variation across study sites.

Table 1. Implementation of Xpert MTB/XDR testing in four study sites[*].

| Study implementer | Country | Xpert-XDR sample collection | Culture/DST |
|---|---|---|---|
| IHVN | Nigeria | Fresh sample taken for Xpert-XDR if MTB detected using Xpert MTB/RIF (regardless of RR) | On-site in three health facilities, at national reference laboratory for one health facility |
| KNCV/ EPHI | Ethiopia | Xpert-XDR done on same sample if MTB detected by Xpert MTB/RIF Ultra (regardless of RR). If insufficient sample left for Xpert-XDR, a fresh sputum sample was requested | At national reference laboratory |
| Wits | South Africa | Fresh sample taken for Xpert-XDR if MTB detected using Xpert MTB/RIF Ultra (regardless of RR) | At provincial (regional) laboratory in same city as the health facility |
| Caprisa | South Africa | Xpert-XDR done on same sample if MTB detected by Xpert MTB/RIF Ultra (regardless of RR). If insufficient sample left for Xpert-XDR, a fresh sputum sample was requested | At provincial (regional) laboratory in same city as the health facility |

[*]Abbreviations used in table: Caprisa – Centre for the AIDS Programme of Research In South Africa; EPHI – Ethiopian Public Health Institute, IHVN – Institute of Human Virology Nigeria, KNCV – KNCV Tuberculosis Foundation, MTB – *Mycobacterium tuberculosis* complex, RIF – rifampicin, RR – rifampicin resistance, Wits – Witwatersrand University, Xpert-XDR – GeneXpert MTB/XDR assay.

## Sampling and recruitment

IDIs were conducted with clinicians, nurses, and laboratory staff at each site. Additionally, we interviewed a small number of policymakers and PWTB in each country to provide additional insight and perspective on the relevance and implementation of a new diagnostic method. We followed the consolidated criteria for reporting qualitative research (COREQ) [15].

No sample size calculations were used in this study as our aim was to reach thematic saturation in interviews with healthcare workers (clinicians, nurses, laboratory staff). In addition, we interviewed PWTB and policy makers to provide additional insight into the impact of the introduction of Xpert-XDR. The intent here was supplemental and not to reach thematic saturation. To determine thematic saturation, weekly meetings were held with all interviewers and qualitative research lead. This ensured interview guides were adapted as needed and to assess for the aforementioned thematic saturation.

## Data collection

Five interviewers (authors: MS, MZ, SII, GG, MB, three females, two males) and one (female) data collector conducted the IDIs between July and November 2023 (11 July – 11 October in sites of the University of the Witwatersrand, 17 July – 27 October in Nigeria, 4 August – 30 September in Ethiopia, and 7 August – 8 November in sites associated with Caprisa). All interviews were conducted in the language the participant was most comfortable in.

Prior to conducting the IDIs, the data collector and interviewers were trained on qualitative research methods, (DR) TB and its diagnosis, and on the TRiAD project. The interviewers varied in qualitative research experience, those with less qualitative experience were very experienced in the TB clinical care pathway. The training included both didactic and practical sessions in which trainees interviewed study staff members with clinical and laboratory backgrounds.

Two interview guides were developed for healthcare workers: one for nurses/clinicians, and one for laboratory staff. The guide for laboratory staff collected insights into the feasibility and acceptability of the GeneXpert instrument and of conducting the Xpert-XDR assay. The guide for nurses/clinicians collected information about the feasibility and acceptability of the new workflow. Additionally, separate interview guides were created for PWTB and policymakers. The themes of the interview guides were organized by the theoretical framework of acceptability [14]. These interview guides were pilot tested with TRiAD study team members during the training in Ethiopia. The interview guides are available in S1 File.

## Data management

Audio recordings, transcriptions and detailed notes were all stored in a secure location. Unique identifiers were used for each interview. All interviews were de-identified prior to analysis; identifying information was removed during the detailed note taking/transcription process.

## Data analysis and reporting

All interviews were audio recorded. In Nigeria, professional transcribers produced verbatim transcripts of interviews. In Ethiopia and South Africa, the interviewers created detailed notes of the interviews. In this process, post interview, the interviewers listened to the audio recordings and took detailed notes, summarizing participant's response to each question.

Data was analysed using a thematic approach, applying both deductive and inductive coding methods. We first started with the deductive coding guided by the themes in the interview guide. A coding scheme was created and agreed upon by all researchers. Weekly research meetings were held with the research team to build upon the coding framework, allowing for new themes to emerge (inductive method). Summaries of emergent themes were created by the researchers in all the participating countries and relevant quotes were collected. Transcripts were coded using qualitative research analysis software, NVivo 12 and NVivo 14 (Lumivero, Denver, USA).

## Ethical considerations

The parent TriAD study was approved by in-country regulatory agencies, and registered on clinic trials.gov, registration number: NCT05175794. Ethical approval for the feasibility and acceptability sub-study was granted in each country: in Ethiopia with reference number EPHI-IRB-384–2021; in Nigeria with reference number NHREC/01/01/2007-01/06/2023; in South Africa for Caprisa with reference number UKZN BREC/00002654/2021; and for Witwatersrand University with reference number HREC: 210805. All interviewees provided written informed consent, including audiotaping, by signing a paper informed consent form prior to the start of the interview.

## Results

A total of 81 interviews were conducted: 61 with healthcare workers, 11 with PWTB, and 9 with policy makers (Table 2). Table 2 provides information on role and demographics of all participants, by country.

## Contributing factors to acceptability

**Time to diagnosis.**  Laboratory staff and healthcare workers found the new workflow feasible and acceptable. They identified the faster turnaround time of genotypic DST result from the Xpert-XDR compared to the conventional pDST as the most significant benefit. This was especially true for the laboratory staff who identified many benefits of the faster turnaround time, including that the faster time paired with a correct resistance pattern reduced the time and need for the duplicative work of re-running individual samples at a later date.

*In previous times, it took 2–3 months to wait for a culture result. In the new machine [Xpert-XDR] the laboratory technician can release the results fast within few hours and send the result to clinician (Lab technician, Ethiopia)*

*The lab technician can then give the clinician the results early, and patients are started on the right treatment immediately. She feels it is beneficial for everyone. (Lab technician, South Africa)*

**Appropriate treatment regimen.**  Prior to implementation of the Xpert-XDR-based algorithm, it may have taken weeks before pDST results became available to guide appropriate treatment. For all participants, having the correct treatment plan, including optimal treatment duration, at time of diagnosis was a major benefit of the Xpert-XDR assay.

*It has been so helpful to us because it makes treatment faster, you can be able to know your patients and know what to give to them at which treatment level. As the patient comes in, then 24–72 hours later, they have commenced treatment, so the rate of spread is limited. (Clinician, Nigeria)*

**Table 2.  Number of interviewees by country, sex and role.**

|  | Laboratory Staff | Clinicians and Nurses | PWTB* | Policy makers |
|---|---|---|---|---|
| **Country** |  |  |  |  |
| *South Africa* | 7 | 13 | 3 | 3 |
| *Nigeria* | 12 | 8 | 4 | 3 |
| *Ethiopia* | 8 | 13 | 4 | 3 |
| **Sex** |  |  |  |  |
| *Male* | 9 | 15 | 4 | 7 |
| *Female* | 18 | 19 | 7 | 2 |

*people with tuberculosis

*After the emergence of the rapid triaging, [we can] detect core drug resistance such as second line injectable-, ethionamide-, and isoniazid- and rifampicin-resistant TB easily. As a result of that, the health professionals firmly rely on the result and decide on the regimen. In general terms, it expands diagnostic services. (Nurse, Ethiopia)*

**Impact on PWTB.** While the main participants in this study were the healthcare workers themselves, we interviewed a number of PWTB to understand their lived experiences while waiting for diagnosis, specifically accurate diagnosis for drug resistance. In some countries, if the DST results indicated resistance, the clinician, after consultation with the DR-TB consilium (or clinic advisory board), would need to change such a person's regimen.

*I was using the short regimen of medicine, that of 9 months. I think the first culture result takes a longer duration for the result to be known, when the result came, I was informed that I had additional resistance to pyrazinamide on top of [the] rifampicin and isoniazid resistance [that] I had at the start. So, they changed the treatment to a longer one with a duration of a year and 8 months. It has been about two months since I started the long regime. (PWTB, Ethiopia)*

As Xpert-XDR results are available sooner than LPA and pDST results, there is the potential for a large impact on the PWTB themselves. Individuals who had their sputum tested during this study were both able to receive their results faster and be placed on the appropriate treatment regimen faster.

*Imagine a person comes with GeneXpert Rif resistance. I start them on a basic long regimen, then they have been on [that] long regimen for four weeks and are pre-XDR. I change the regimen and another four weeks later I find bedaquiline resistance. Now I have compromised the regimen, now they are going to a rescue regimen. We have started two months of inappropriate regimens and have subjected this person to 6–18 months of inpatient treatment, injecting them every day. So, it has been a pretty difficult process and a painful process for the patient. (Clinician, South Africa)*

## Contributing factors to feasibility

**Xpert-XDR implementation.** Feasibility of implementation of Xpert-XDR testing was examined through the lens of healthcare facility implementation. As the Xpert-XDR assay is done using the same processing procedures and the same (but updated) instrument as the Xpert MTB/Rif (Ultra) assay, and thus implies a variation of existing workflow algorithms, communication pathways remain unchanged.

**Training.** Training was provided to nearly all laboratory staff by the TRiAD study team. Laboratory staff at all sites found this training satisfactory as it met their needs. Furthermore, the training on conduct of the assay and use of the instrument was familiar for many laboratory technicians as the basic procedures conducted for the Xpert MTB/Rif (Ultra) assay were similar to the Xpert-XDR assay.

*The training was easy because [of] the [Xpert MTB/]Rif [and Xpert] Ultra; [it] was the same procedure, nothing has changed, it is just is more about melt curves and temperature, it uses different methods for the assay. It is giving you more drugs [susceptibility results] in a broader spectrum, but the whole preparation of the test is all the same. (Lab technician, South Africa)*

However, training varied across the levels of healthcare workers and sites. Nurses, for example, had contrasting training experiences in the three different countries; from no training in Ethiopia to all nurses being training at a site in South Africa. A clinician in Ethiopia highlighted some of the challenges that can come with training specific staff involved in a study instead of clinic-wide training.

*The participation of staff in cohort studies like this is limited even at the national level. So, the level of understanding among different healthcare staff is not similar. Therefore, on [the] job assistance might be required. (Clinician, Ethiopia)*

In the case of implementation of the Xpert-XDR assay, role-appropriate and on-going trainings were recognized as essential considerations in ensuring that new diagnostic workflows are appropriately followed, and knowledge of the diagnostic tool is shared appropriately.

*What's really needed is ongoing training because you get to train people, they change all the time. You know you find new MDR doctors; all the time they are new [...] so ongoing training is critical both at National Health Laboratory Service and within their facilities. (Policy maker, South Africa)*

### Barriers to feasibility & acceptability

Despite the strengths, some barriers were identified. Laboratory staff expressed concern that the Xpert-XDR assay has a lower sensitivity for TB detection when compared to Xpert-Ultra. This led occasionally to discordant results (false negative on Xpert XDR).

*All these individual patients must have [Xpert-]Ultra resolved first before it's eligible for 10c [Xpert-XDR], so we discovered that this [Xpert-]Ultra is more sensitive than 10c, the sensitivity is low, that is the basic technical challenge we have, the sensitivity is low compared to [Xpert-] Ultra […] when you detect with [Xpert-]Ultra as MTB detected, RIF resistant, it could still be eligible because you want to check for isoniazid resistance, so because of that, sometimes even using that same sample on 10color, you can still get it negative. (Lab technician, Nigeria)*

Participants described how the discordant results could disrupt workflow, particularly affecting laboratory staff, as samples would either need to be rerun or new samples would need to be collected. Participants highlighted that individuals with presumptive TB may need to produce a new sputum specimen, which can be challenging since these samples must be collected at least one hour apart. Consequently, they might need to return on a different day to provide the specimen.

The Xpert-XDR assay is run on a GeneXpert instrument, which is power dependent. Participants from all countries identified unreliable power as a major barrier. In South Africa, this took the form of nationwide power cuts or load shedding, which can be, but is not always, on a schedule. In Ethiopia, laboratory staff also identified unstable internet connectivity as a barrier. In Nigeria, laboratory technicians mentioned the impact of managing temperature control in the room where the machines are placed. Lab technicians in both Ethiopia and South Africa mentioned the importance of uninterruptable power supply to mitigate challenges. In Nigeria, solar power was used to offset electrical power.

*For these kind of instruments, you need to have a UPS [uninterruptable power supply] while Xpert-XDR is running because the instrument will not know at what point in the run it stopped just before power cuts and sometimes you would never know because load shedding can happen anytime and if you have [a] sample to run, you cannot wait to run it after load shedding. Having a UPS helps. (Lab technician, South Africa)*

Power interruptions can also impact on the supplies, as cartridges become non-functional once the power is interrupted while the test is running. The financial impact of lost supplies was identified as a major barrier to implementation by a health-decision maker.

*The Xpert-XDR is highly power sensitive and once the power is interrupted while the test is being done, we cannot get the result, and we need to repeat the test with another cartridge which is resource intensive. There is also no power*

*in many primary health facilities and that may also affect what is planned in the strategic plan. One test costs $20 and missing one cartridge without a result means a big loss. With a one-time interruption due to the power interruption, 4 cartridges can get lost. (Policy Maker, Ethiopia)*

The heat sensitivity of the GeneXpert instrument is another significant constraint for feasibility of Xpert testing. The instrument's electronic components are prone to overheating, which can lead to damage and malfunctioning of the equipment.

*The challenge is that [the] machine is heat sensitive and because of their small nature they can easily be heated up, and once they are heated up, they begin to misbehave. The challenge also is sometimes module failure, and it could be traced to the issue of temperature. (Lab technician, Nigeria)*

### Requirements for implementation

Healthcare decision-makers also spoke to the importance of ensuring support across all levels of the health service for implementing changes to national guidelines incorporating Xpert-XDR testing into standard of care. This highlighted the importance of ensuring that guidelines provided to clinics are clear and concise and that information flows to all necessary role-players.

*We must make sure that all our policy guidelines capture these tools and have clear algorithms [regarding Xpert-XDR]. It must be declared in our algorithm and then when we have it there in our policy document, in our manuals, in our SOPs, then we must make sure people are well informed in terms of training. (Policy maker, Ethiopia)*

## Discussion

This study revealed that healthcare workers found implementation of the new Xpert-XDR assay both feasible and acceptable, with the rapid turnaround time being its most significant advantage. The multiple resistance patterns that the Xpert-XDR assay is able to identify supports correct diagnosis and guidance on appropriate treatment regimens using a simple algorithm; in which Xpert-Ultra is the initial diagnostic test, followed by Xpert-XDR. This increases the possibility of a rapid diagnosis of MDR- and pre-XDR-TB and reduces the need for a PWTB to return for treatment corrections. At a clinic-level, the faster turnaround time enables the clinician to place the PWTB on the appropriate treatment regimen from the initial appointment. This reduces cost and effort required for repeat appointments by PWTB, as well as repeated work by the healthcare team in addition to improving the treatment process for PWTB.

As advanced technology becomes more widely used at subnational levels, lessons learned should be considered when designing future implementation plans. For instance, the deployment of Xpert-XDR testing should be accompanied by a carefully designed and adequately funded plan for maintenance and infrastructure, including power supply and ensuring a consistent supply chain. The development of alternative solutions, such as improved thermal management systems and long-lasting modules, could also improve the machine's sustainability and cost-effectiveness in resource-constrained settings [16].

Healthcare facilities and diagnostic centres need to invest not only in the equipment and supplies, but also in the healthcare workers themselves. With constant turnover and different training requirements per role, it is essential that regular trainings be included in future implementation. Trainings need to be integrated into standard operating procedures at health facilities. These trainings also need to be role appropriate. For example, insufficient training in sputum processing for Xpert testing may produce results with higher-than-acceptable rates of error, as shown in several studies [17].

The Xpert MTB/XDR assay has a sensitivity of 94–100% and a specificity of 100% for all drugs except for ethionamide in comparison to sequencing. This performance is in the same range as that of pDST [18,19]. With the Xpert-XDR assay, used as a reflex test or in parallel, a rapid diagnostic algorithm can now be available onsite. This enables the assay to be used in peripheral sectors of the global health care system, where more rapid identification of expanded drug

resistance may improve therapeutic decision-making, specifically by clinicians. Thus Xpert-XDR is best used after TB detection by Xpert-Ultra or similar assay. However, a challenge identified by the laboratory technicians was the reduced sensitivity of Xpert-XDR in TB detection compared to Xpert-Ultra. In paucibacillary specimens, Xpert-Ultra may detect (rifampicin-resistant) MTB, while Xpert-XDR detects no MTB [20,21].

A major limitation of the Xpert-XDR assay is its inability to detect XDR-TB following the current definition of MDR-TB with additional resistance to any fluoroquinolone and another WHO group A drug (bedaquiline or linezolid) [3]. While Xpert-XDR can detect resistance to isoniazid and fluoroquinolones, it cannot assess resistance to linezolid and/or bedaquiline. Although this was not mentioned as a limitation by the interviewees, multiple countries having a sufficient level of laboratory expertise and infrastructure have started replacing Xpert XDR by sequencing-based diagnostics [22]. Newer diagnostic tests based on whole genome or targeted next generation sequencing offer more flexibility to include new resistance patterns than ready-to-use "closed" assays, such as Xpert-XDR, although they are more complex and more expensive [23]. However, as sequencing based assays are currently beyond the expertise of many countries, it is expected that one-stop-shop tests such as the Xpert MTB/RIF (Ultra), and Xpert-XDR will remain important for the detection of TB and resistance additional to rifampicin.

All of the other barriers identified, including temperature control and power outages, are not specific to the Xpert-XDR assay. Rather, these are related to the GeneXpert instrument and other power dependent diagnostic equipment, which requires stable electricity during operation, and stable temperature during test operation and storage of cartridges. This includes other low-complexity nucleic acid amplification tests recommended by the WHO [5]. Provision of stable electricity can be overcome by using batteries and/or solar energy. In India, the installation of solar panels for stable power in two remote sites brought the power-outage related error rate of Xpert MTB/RIF down from 2.1% to 0.9% [16].

## Limitations

There were several limitations to this study. First, there was great variation among study sites in how the sputum for testing on Xpert-XDR was collected (Table 1). This impacted the authors' ability to understand time to appropriate treatment initiation across study locations. When implementing this assay, the number of sputum samples to be collected should be considered from a patient-centred approach, limiting the times they need to come to a site to produce sputum.

Second, all trainings on Xpert-XDR were conducted by the in-country TRiAD study team as part of an implementation trial [13]. At the time of our study, these trainings were not yet part and parcel of the routinely implemented continuous National Tuberculosis Program (NTP) training programmes as the Xpert-XDR assay was not yet part of the TB diagnostic algorithms in all of the study settings. Staff interviewed in this study may thus have had different experiences compared to staff at other sites or those to be interviewed in the in the future.

## Conclusion

Rapid triaging by Xpert MTB/Rif (Ultra), followed by Xpert-XDR has demonstrated feasibility and acceptability among healthcare workers, primarily due to its rapid turnaround time for DR-TB diagnosis. This advantage is crucial as it supports prompt identification of multiple resistance patterns, facilitating the correct diagnosis, and appropriate and early treatment of TB. Using this rapid triaging algorithm, healthcare providers can rapidly diagnose rifampicin- and multidrug- resistant TB as well as fluoroquinolone-resistant MDR-TB, minimizing the need for patients to return for treatment adjustments due to poly-resistance. Health system wide support in terms of training and machine maintenance, along with considerations such as stable power are essential in supporting the sustained implementation of Xpert-XDR at the subnational level. By identifying and addressing the barriers associated with the instrument, healthcare professionals and policymakers can facilitate better tuberculosis diagnosis, treatment, and control, ultimately improving healthcare outcomes. Finally, it should be noted that additional tests, such as pDST or (targeted) genome sequencing are needed to detect resistance to key drugs as bedaquiline and linezolid. Therefore, it is crucial to assess turnaround times and cost-effectiveness of combining Xpert-XDR which such tests, especially for pre-XDR-TB patients.

## Supporting information

**S1 File. Interview topic guides for laboratory technicians, nurses/clinicians, patients and policy makers used during the in-depth interviews.**
(DOCX)

## Acknowledgments

This publication was produced by the TRiAD study. We express our gratitude to all study participants, data collectors (particularly Sonia Goliath) as well as interview transcribers and translators. Ineke Spruijt and Liza de Groot are thanked for preparing the protocol for this study.

The TRiAD consortium is made up of partners from the following institutions: Centre for the Aids Programme of Research in South Africa, Durban, KwaZulu-Natal, South Africa; SAMRC-CAPRISA HIV-TB Pathogenesis and Treatment Research Unit, Doris Duke Medical Research Institute, University of KwaZulu-Natal, Durban, KwaZulu-Natal, South Africa; Institute of Human Virology Nigeria, Abuja, Nigeria; Clinical HIV Research Unit (CHRU), WITS Health Consortium, Jose Pearson TB Hospital, Bethelsdorp, Port Elizabeth, South Africa; Koninklijke Nederlandse Centrale Vereniging tot Bestrijding der Tuberculose, The Hague, The Netherlands; Koninklijke Nederlandse Centrale Vereniging tot Bestrijding der Tuberculose, Addis Ababa, Ethiopia; Amsterdam Institute for Global Health and Development, Amsterdam, North Holland, The Netherlands; The University of St Andrews, St Andrews, The United Kingdom of Great Britain and Northern Ireland; Foundation for Innovative New Diagnostics, Genève, Geneva, Switzerland; Ethiopian Public Health Institute, Addis Ababa, Ethiopia; IRCCS Ospedale San Raffaele, Milano, Italy; Global Alliance for TB Drug Development, New York, USA; and National Institute for Medical Research, Dar es Salaam, Tanzania.

## Author contributions

**Conceptualization:** Sara Keller, Kogieleum Naidoo, Edine W. Tiemersma.

**Data curation:** Sara Keller, Medhane Zekarias, Sussan Israel-Isah, Mohammed Shaka, Gcinile Gule, Miriam Bathnna, Edine W. Tiemersma.

**Formal analysis:** Sara Keller, Medhane Zekarias, Sussan Israel-Isah, Mohammed Shaka, Gcinile Gule, Miriam Bathnna.

**Funding acquisition:** Kogieleum Naidoo, Anushka Naidoo, Edine W. Tiemersma.

**Investigation:** Sara Keller, Medhane Zekarias, Sussan Israel-Isah, Mohammed Shaka, Gcinile Gule, Miriam Bathnna, Edine W. Tiemersma.

**Methodology:** Sara Keller, Medhane Zekarias, Sussan Israel-Isah, Mohammed Shaka, Gcinile Gule, Miriam Bathnna.

**Project administration:** Sara Keller, Kogieleum Naidoo, Anushka Naidoo, Edine W. Tiemersma.

**Resources:** Sara Keller, Edine W. Tiemersma.

**Supervision:** Sara Keller, Edine W. Tiemersma.

**Validation:** Sara Keller, Medhane Zekarias, Sussan Israel-Isah, Mohammed Shaka, Gcinile Gule, Miriam Bathnna, Edine W. Tiemersma.

**Writing – original draft:** Sara Keller, Medhane Zekarias, Sussan Israel-Isah, Mohammed Shaka, Gcinile Gule, Miriam Bathnna, Edine W. Tiemersma.

**Writing – review & editing:** Sara Keller, Kogieleum Naidoo, Medhane Zekarias, Sussan Israel-Isah, Mohammed Shaka, Gcinile Gule, Anushka Naidoo, Miriam Bathnna, Judith Nomthandazo Dlamini-Miti, Kalkidan Yae, Evaezi Okpokoro, Alash'le Abimiku, Ahmed Bedru, Edine W. Tiemersma.

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
