## [Decision Letter · Decision Letter 0]

PONE-D-24-58843Feasibility and acceptability of GeneXpert MTB/XDR workflow implementation among healthcare workers in three low-middle income African countriesPLOS ONE

Dear Dr. Tiemersma,

Thank you for submitting your manuscript to PLOS ONE. After careful consideration, we feel that it has merit but does not fully meet PLOS ONE’s publication criteria as it currently stands. Therefore, we invite you to submit a revised version of the manuscript that addresses the points raised during the review process.

Please see below the reviewers comments and I hope you find it helpful to further improve your manuscript. Please address the reviewers concerns carefully and I encourage you to please include that in your manuscript, not just write an answer in your response to reviewers. Especially, please consider reviewer 1 recommendation on the title. 

We look forward to receiving your revised manuscript.

Kind regards,

Padmapriya P Banada, PhD

Academic Editor

PLOS ONE

3. Thank you for stating the following financial disclosure:  [This publication was produced by the TRiAD study which is part of the EDCTP2 program supported by the European Union (Grant RIA2019-2888-TRIAD).].  Please state what role the funders took in the study.  If the funders had no role, please state: "The funders had no role in study design, data collection and analysis, decision to publish, or preparation of the manuscript." If this statement is not correct you must amend it as needed. Please include this amended Role of Funder statement in your cover letter; we will change the online submission form on your behalf.

Please update your Data Availability statement in the submission form accordingly**.**

Additional Editor Comments:

Please see below the reviewers comments and I hope you find it helpful to further improve your manuscript. Please address the reviewers concerns carefully and I encourage you to please include that in your manuscript, not just write an answer in your response to reviewers. Especially, please consider reviewer 1 recommendation on modifying the title.

Reviewers' comments:

Reviewer's Responses to Questions

**Comments to the Author**

1. Is the manuscript technically sound, and do the data support the conclusions?

Reviewer #1: Yes

Reviewer #2: Yes

2. Has the statistical analysis been performed appropriately and rigorously? 

Reviewer #1: Yes

Reviewer #2: N/A

3. Have the authors made all data underlying the findings in their manuscript fully available?

Reviewer #1: Yes

Reviewer #2: No

4. Is the manuscript presented in an intelligible fashion and written in standard English?

Reviewer #1: Yes

Reviewer #2: Yes

5. Review Comments to the Author

Reviewer #1: I have reviewed the manuscript by Keller et al. The manuscript is addressing a relevant aspect of the diagnosis of drug resistant TB using the GeneXpert XDR test. It is reporting the Feasibility and acceptability of the test workflow.

I have raised comments to be addressed below:

Major comment

1) The manuscript is about workflow, how ever the content is mainly about acceptability of the test and less of the workflow. The authors should have looked into the proposed workflow of the GeneXpert XDR and conduct a study looking into the feasibility and acceptability of such steps towards GeneXpert XDR testing. For examples, the feasibility and acceptability of testing the same sample as GeneXpert Ultra Vs collecting another sample, on the same day versus on a different day. looking into both lab, clinic and patient workflow. This could be still looked at through the same study population, Lab, clinician and PWTB. If this is not the case, then, word "Workflow" should be removed from the title

and the entire writeup of the manuscript revised to concentrate on the "Feasibility and Acceptability of the test only.

Minor comments

1) abstract: L38: the why "especially laboratories" yet the manuscript is about lab personnel, clinician patients

2) The study happened in only 4-months, was this nested in another already existing study or it was the main and primary study? this seems to be a very short time for a multi-country study.

3) There is less literature about the concepts of the study under L108-112

4) L132-133: were these numbers considered adequate? Usually IDs consider a sample size of 9-17 participants to reach saturation.

5) L143: Very difficult to find interviewers who can speak all languages the participant feels comfortable! may be the language the can understand.

6) L146: The interviewers varied in different strength, BUT the study had only 5 interviewers

7) Results section: The number of people interviewed are higher than those intended as per L132: Did the REC approve the additional numbers of interviewees than those intended for?

8) L188-190: Is this section for methods or these are results?

9) L209: This should have a sub section i.e "Factors contributed to Acceptability" and avoid repeating "acceptability or Feasibility going forward.

10) L290: "Lower sensitivity for TB detection". Since this study was not a diagnostic accuracy study, there is possibility of this assumption not being true. There are several factors such as sample collection, bacterial load in the sample and processing as you indicate in table 2 that may affect the outcome of the test. I suggest you change to "discordant results" and not lower sensitivity.

Reviewer #2: The manuscript is well-written.

The authors have stated that the data cannot be publicly shared as it consists of individual opinions provided under the condition of deidentification and restricted access to the research team. However, they could consider sharing a checklist of interview questions along with the aggregated qualitative responses in numerical or percentage form.

The GeneXpert MTB/XDR assay is unable to detect resistance to key drugs such as linezolid, bedaquiline (Group A drugs under the new WHO guidelines), pyrazinamide (PZA), and other repurposed second-line drugs. Furthermore, targeted NGS from clinical specimens is not yet feasible in many low- and middle-income countries (LMICs) with a high burden of DR-TB, making phenotypic DST (pDST) or molecular PCR assays essential for detecting susceptibility or resistance to these drugs. Therefore, it is crucial to assess the turnaround time and cost-effectiveness of using GeneXpert MTB/XDR in combination with pDST or additional molecular PCR assays for drugs like PZA, bedaquiline, and linezolid in XDR-TB patients.

6. PLOS authors have the option to publish the peer review history of their article (what does this mean? ). If published, this will include your full peer review and any attached files.

**Do you want your identity to be public for this peer review?** For information about this choice, including consent withdrawal, please see our Privacy Policy .

Reviewer #1: **Yes: ** Willy Ssengooba

Reviewer #2: No

---

## [Author Response · Author response to Decision Letter 1]

11 Apr 2025

Review Comments to the Author

Reviewer #1:

I have reviewed the manuscript by Keller et al. The manuscript is addressing a relevant aspect of the diagnosis of drug resistant TB using the GeneXpert XDR test. It is reporting the Feasibility and acceptability of the test workflow.

I have raised comments to be addressed below:

Major comment

1) The manuscript is about workflow, how ever the content is mainly about acceptability of the test and less of the workflow. The authors should have looked into the proposed workflow of the GeneXpert XDR and conduct a study looking into the feasibility and acceptability of such steps towards GeneXpert XDR testing. For examples, the feasibility and acceptability of testing the same sample as GeneXpert Ultra Vs collecting another sample, on the same day versus on a different day. looking into both lab, clinic and patient workflow. This could be still looked at through the same study population, Lab, clinician and PWTB. If this is not the case, then, word "Workflow" should be removed from the title and the entire writeup of the manuscript revised to concentrate on the "Feasibility and Acceptability of the test only.

Reply: We thank the reviewer for this comment and agree that this paper is about the overall implementation of Xpert MTB/XDR and the impact of adding this new test on the work of health care workers. While we, indeed, did not look specifically and separately at each step in the workflow, we did intend to broadly examine clinic workflow. To address the reviewer’s comment, we have removed the word ‘workflow’ from the title and at places in the manuscript where we felt that the term was not necessary to keep the meaning of the sentence. However, where discussing the overall broader impact of the new workflow, we left the term in.

Minor comments

1) abstract: L38: the why "especially laboratories" yet the manuscript is about lab personnel, clinician patients

Reply: This line concerns background information, and we believe that introduction of the Xpert MTB/XDR test will especially change the workflow in those laboratories that previously did not conduct Xpert testing, as this means a new test being introduced with an additional follow-on (reflex) test that needs to be done and interpreted, and additional results to be shared with clinicians, whereas previously, another sample might have been collected for submission to a reference laboratory.

2) The study happened in only 4-months, was this nested in another already existing study or it was the main and primary study? this seems to be a very short time for a multi-country study.

Reply: This study was indeed nested into a larger study, as is briefly explained in the abstract (lines 43-44) and in more detail in lines 108-110. The data collection could be done in just 4 months as data were concurrently collected by different teams in each country (2 interviewers per team).

3) There is less literature about the concepts of the study under L108-112

Reply: Indeed, this paragraph does not contain any references to literature as it pertains to the study itself, explaining the aims of our study. Therefore, no references were added.

4) L132-133: were these numbers considered adequate? Usually IDs consider a sample size of 9-17 participants to reach saturation.

Reply: The main population of our study were health care workers. Interviews with patients and policy makers were added to enrich our data. We admit that this may not have become very clear from the manuscript. Therefore, we changed lines 132-136 as follows: “...our aim was to reach thematic saturation in interviews with healthcare workers (clinicians, nurses, laboratory staff). In addition, we interviewed PWTB and policy makers to provide additional insight into the impact of the introduction of Xpert-XDR. The intent here was supplemental and not to reach thematic saturation.”

5) L143: Very difficult to find interviewers who can speak all languages the participant feels comfortable! may be the language the can understand.

Reply: Interviewers were hired on the basis that they could speak the languages of the interviewees. So indeed, they did feel comfortable in the languages used during the interview.

6) L146: The interviewers varied in different strength, BUT the study had only 5 interviewers

Reply: As explained in lines 141-142, there were six persons conducting the interviews (five interviewers and one data collector). As explained in the next paragraph, all these were trained both in TB and in qualitative data collection skills. During the training, trainees with less experience in TB but strong qualitative interviewing skills were paired with trainees with more experience in TB but less experience in conducting qualitative interviews.

7) Results section: The number of people interviewed are higher than those intended as per L132: Did the REC approve the additional numbers of interviewees than those intended for?

Reply: In response to comment #4 of this reviewer, the number of intended interviewees has been removed from the methods section as numbers of interviews planned concerned the maximum number of interviews considered to be needed to reach thematic situation, and are thus not essential. The total number of interviews was lower than this maximum in all countries (25 health care workers, 4 persons with TB and 3 policy makers per country): 20, 20 and 21 health care workers, 3 ,4 and 4 PWTB, and 3, 3 and 3 policy makers were interviewed respectively in South Africa, Nigeria and Ethiopia. Final numbers of interviews conducted are listed in Table 2 in the Results section.

In direct response to the comment of going above numbers - there was no need for approval as in qualitative research there is not a strict number of interviews required, but rather the aim of reaching thematic saturation. See also our response to comment #4.

8) L188-190: Is this section for methods or these are results?

Reply: The sentences “The laboratory staff provided insights into the feasibility and acceptability of the GeneXpert instrument and of conducting the Xpert-XDR assay. The clinicians and nurses provided feedback about the acceptability and feasibility of the new workflow” rather describe the type of information to be collected in the interviews than the precise information that was obtained. We have now changed the sentences slightly and moved them to the methods section (lines 154-156) as proposed by the reviewer.

9) L209: This should have a sub section i.e "Factors contributed to Acceptability" and avoid repeating "acceptability or Feasibility going forward.

Reply: Subheadings have been adapted to reflect this comment of the reviewer.

10) L290: "Lower sensitivity for TB detection". Since this study was not a diagnostic accuracy study, there is possibility of this assumption not being true. There are several factors such as sample collection, bacterial load in the sample and processing as you indicate in table 2 that may affect the outcome of the test. I suggest you change to "discordant results" and not lower sensitivity.

Reply: While we agree with the reviewer that this is not necessarily true, the text describes results directly associated with the responses from the participants, and not our assumptions. Thus, we feel that leaving the wording as is, is most true to the results.

Reviewer #2:

The manuscript is well-written.

The authors have stated that the data cannot be publicly shared as it consists of individual opinions provided under the condition of deidentification and restricted access to the research team. However, they could consider sharing a checklist of interview questions along with the aggregated qualitative responses in numerical or percentage form.

Reply: While the reviewer provides an interesting suggestion which we will consider if the editor thinks this is needed, we considered this not feasible as the first author has left KNCV and no longer has direct access to the database. Instead, we have added all interview guides to the manuscript as a supplemental file (see reference to this in line 160).

The GeneXpert MTB/XDR assay is unable to detect resistance to key drugs such as linezolid, bedaquiline (Group A drugs under the new WHO guidelines), pyrazinamide (PZA), and other repurposed second-line drugs. Furthermore, targeted NGS from clinical specimens is not yet feasible in many low- and middle-income countries (LMICs) with a high burden of DR-TB, making phenotypic DST (pDST) or molecular PCR assays essential for detecting susceptibility or resistance to these drugs. Therefore, it is crucial to assess the turnaround time and cost-effectiveness of using GeneXpert MTB/XDR in combination with pDST or additional molecular PCR assays for drugs like PZA, bedaquiline, and linezolid in XDR-TB patients.

Reply: We agree with the reviewer that this is a major limitation of the Xpert-XDR assay, although it was not raised as a limitation by the interviewees. We therefore included this as a limitation in the Discussion section (lines 398-407). To strengthen the manuscript, we now added to the concluding section (lines 443-447): “Finally, it should be noted that additional tests, such as pDST or (targeted) genome sequencing are needed to detect resistance to key drugs as bedaquiline and linezolid. Therefore, it is crucial to assess turnaround times and cost-effectiveness of combining Xpert-XDR which such tests, especially for pre-XDR-TB patients.”

---

## [Editor Report · Decision Letter 1]

PONE-D-24-58843R1Feasibility and acceptability of GeneXpert MTB/XDR implementation among healthcare workers in three low-middle income African countriesPLOS ONE

Dear Dr. Tiemersma,

Thank you for submitting your manuscript to PLOS ONE. After careful consideration, we feel that it has merit but does not fully meet PLOS ONE’s publication criteria as it currently stands. Therefore, we invite you to submit a revised version of the manuscript that addresses the points raised during the review process.

Thank you for submitting your revised manuscript and considering the reviewers comments. Although I accept most of your rebuttal, some remain unanswered. for Reviewer 1 question 3 about the reference, I believe the reviewer is intending to ask if your proposed workflow is supported by any other published literature? and if so, please give a reference although we understand this is your protocol. however, if this protocol is completely novel to this study, you will have to explain the need for deviation from similar published studies. As for the reviewer 2's concern about the data sharing, please note that PLOS journals require authors to make all data underlying the findings described in their manuscript fully available without restriction when at all possible. The data should be part of the institution not the author who has left. Please make every effort to make all data available

We look forward to receiving your revised manuscript.

Kind regards,

Padmapriya P Banada, PhD

Academic Editor

PLOS ONE

Journal Requirements:

Additional Editor Comments :

Dear authors,

Thank you for submitting your revised manuscript and considering the reviewers comments.

Although I accept most of your rebuttal, some remain unanswered. for Reviewer 1 question 3 about the reference, I believe the reviewer is intending to ask if your proposed workflow is supported by any other published literature? and if so, please give a reference although we understand this is your protocol. however, if this protocol is completely novel to this study, you will have to explain the need for deviation from similar published studies.

As for the reviewer 2's concern about the data sharing, please note that PLOS journals require authors to make all data underlying the findings described in their manuscript fully available without restriction when at all possible. The data should be part of the institution not the author who has left. Please make every effort to make all data available.

---

## [Author Response · Author response to Decision Letter 2]

14 May 2025

Response to editor’s comments:

1) For Reviewer 1 question 3 about the reference, I believe the reviewer is intending to ask if your proposed workflow is supported by any other published literature? and if so, please give a reference although we understand this is your protocol. however, if this protocol is completely novel to this study, you will have to explain the need for deviation from similar published studies.

We apologize the comment of the reviewer. Re-reading the comment (“There is less literature about the concepts of the study under L108-112”) as well as the respective paragraph, we understand that the reviewer wants to know if there is any literature underlying the theoretical framework of acceptability and feasibility tha. The theoretical framework of feasibility and acceptability for this study was based on work published by Sekhon et al. (reference 14 in the original manuscript).

With regards to the introduction of the Xpert MTB/XDR test in the laboratory workflow, this is based on recommendations by the manufacturer. This is further explained in a publication of the TRiAD study protocol (which was added to the original manuscript as reference 23).

We have added references to both papers in lines 108-114 as follows: “The TriAD study aims to evaluate the effectiveness, feasibility, acceptability and cost-effectiveness of implementing a rapid triaging algorithm with the use of Xpert-XDR and selection of all-oral regimens for DR-TB in South Africa, Ethiopia, and Nigeria (13). To assess the programmatic and operational aspects of introducing this algorithm, we assessed its feasibility and acceptability, using the theoretical framework of acceptability of healthcare interventions developed by Sekhon et al. (14). We included barriers and facilitators from the perspective of healthcare workers, policy makers and PWTB in South Africa, Ethiopia, and Nigeria.”

All references have been renumbered accordingly.

2) As for the reviewer 2's concern about the data sharing, please note that PLOS journals require authors to make all data underlying the findings described in their manuscript fully available without restriction when at all possible. The data should be part of the institution not the author who has left. Please make every effort to make all data available.

We do apologize for our confusing answer; as we failed to update it in line with the text we added to the Data Availability Statement. Here, we do explain that the data is available on request and that it can be made available by addressing the respective ethical review boards: “The qualitative data collected for this study contains sensitive information that could lead to the identification of patients, staff and policy makers, despite the data having been de-identified. Data access requests can be made:

• For South Africa, Caprisa sites, via the Biomedical Research Ethics Committee of the University of Kwazulu-Natal at the following e-mail address:BREC@ukzn.ac.za, citing protocol reference number BREC/00002654/2021; for the University of Witwatersrand via the Human Research Ethics Committee at the following email address: EthicsRegulatory@witshealth.co.za, citing protocol reference number 210805;

• For Nigeria via the National Health Research Ethics Committee of Nigeria at the following email address: secretary@nhrec.net, citing NHREC approval number NHREC/01/01/2007-01/06/2023;

• For Ethiopia via the Ethiopian Public Health Institute Institutional Review Board at the following email address: info@ephi.gov.et, citing the EPHI-IRB and protocol reference number EPHI-IRB-384-2021”

---

## [Editor Report · Decision Letter 2]

Feasibility and acceptability of GeneXpert MTB/XDR implementation among healthcare workers in three low-middle income African countries

PONE-D-24-58843R2

Dear Dr. Tiemersma,

We’re pleased to inform you that your manuscript has been judged scientifically suitable for publication and will be formally accepted for publication once it meets all outstanding technical requirements.

Kind regards,

Padmapriya P Banada, PhD

Academic Editor

PLOS ONE

Additional Editor Comments (optional):

Thank you for revising the manuscript and responding to all the concerns.
---

## [Editor Report · Acceptance letter]

PONE-D-24-58843R2

PLOS ONE

Dear Dr. Tiemersma,

I'm pleased to inform you that your manuscript has been deemed suitable for publication in PLOS ONE. Congratulations! Your manuscript is now being handed over to our production team.

Kind regards,

on behalf of

Dr. Padmapriya P Banada

Academic Editor

PLOS ONE